

# Collection and analysis of a global marine phytoplankton primary production dataset

Franceso Mattei[1,2,3], Michele Scardi[1,2]

[1]Department of Biology, University of Rome "Tor Vergata", Via della Ricerca Scientifica (no street number), Rome, 00133, Italy.
[2]CoNISMa, Piazzale Flaminio, 9, Rome, 00196, Italy.
[3] Ph.D. Program in Evolutionary Biology and Ecology, Department of Biology, University of Rome Tor Vergata.

*Correspondence to*: Francesco Mattei ([francesco.mattei90@yahoo.it](mailto:francesco.mattei90@yahoo.it))

**Keywords**: Phytoplankton primary production, Marine ecology, Global marine primary productivity.

**Abstract.** Phytoplankton primary production is a key oceanographic process. It has relationships with the marine food webs dynamics, the global carbon cycle and the Earth's climate. The study of phytoplankton production on a global scale relies on indirect approaches due to field campaigns difficulties. Modelling approaches require in situ data for calibration and validation. In fact, the need for more phytoplankton primary production data was highlighted several times during the last decades.

Most of the available primary production datasets are scattered in various repositories, reporting heterogeneous information and missing records. We decided to retrieve field measurements of marine phytoplankton production from several sources and create a homogeneous and ready to use dataset. We handled missing data and added variables related to primary production which were not present in the original datasets. Subsequently, we performed a general analysis of the highlighting the relationships between the variables from a numerical and an ecological perspective.

Data paucity is one of the main issues hindering the comprehension of complex natural processes.

We believe that an updated and improved global dataset, complemented by an analysis of its characteristics, can be of interest to anyone studying marine phytoplankton production and the processes related to it. The dataset described in this work is published in the PANGAEA repository. DOI: https://doi.pangaea.de/10.1594/PANGAEA.932417 (Mattei and Scardi, 2021)

## 1 Introduction

Phytoplankton primary production is a pivotal process in biological oceanography. It accounts for roughly 98% of the marine system autotrophic production and 50% of the global one (Carvalho et al., 2017; Field et al., 1998). Accordingly, this process provides the main source of energy for structuring the marine food webs (Duarte and Cebrián, 1996; Kwak and Park, 2020a). Furthermore, it influences the absorption of carbon dioxide from the atmosphere and the flux of carbon to the deep ocean generating a process known as biological pump (Giering et al., 2014; Longhurst and Glen Harrison, 1989). The estimated




global phytoplankton production is comprised between 30 and 70 Gt C y$^{-1}$ (Carr et al., 2006; Friedrichs et al., 2009; Saba et
        al., 2010; Siegel et al., 2013), i.e. most probably still larger than the global anthropogenic $CO_2$ emissions (roughly 37 Gt $CO_2$
        y$^{-1}$) (Caldeira and Duffy, 2000; Falkowski and Wilson, 1992; Jackson et al., 2019; Peters et al., 2020; Sabine et al., 2004).
        These features highlight the strong link between phytoplankton production and both ecosystem services and Earth's climate
        (Barange et al., 2014; Behrenfeld et al., 2006; Blanchard et al., 2012a; Blythe et al., 2020). This link in turn reflects the central
role of this biological process not only in the oceans' dynamics, but also in those of the whole geo-biosphere.
        The availability of remotely sensed information allowed the study of the phytoplankton production at a global scale providing
        a synoptic view of several ocean features, such as chlorophyll *a* surface concentration, Sea Surface Temperature (SST) and
        Photosynthetic Active Radiation (PAR) (Groom et al., 2019; Platt and Sathyendranath, 1988; Sammartino et al., 2018;
        Westberry and Behrenfeld, 2014). Several models which exploits satellite information to estimate primary production have
been proposed (e.g. Behrenfeld and Falkowski, 1997; Friedrichs et al., 2009; Mattei and Scardi, 2020; Westberry and
        Behrenfeld, 2014). In fact, estimators of this process provide valuable tools to assess global phytoplankton production
        characteristics and patterns which in turn could provide insights into the dynamics of several phenomena, e.g. fishery yields
        and climate change effects (Fox et al., 2020; Richardson and Schoeman, 2004; Russo et al., 2019).
        Nevertheless, the lack of field data negatively affects the power and the reliability of both satellite information and model
estimates. In fact, these data are essential to calibrate satellite sensors and develop primary production estimators.
        The most complete and freely accessible phytoplankton production dataset is available at
        http://sites.science.oregonstate.edu/ocean.productivity/field.data.c14.online.php. From now on we will refer to this data as the
        Ocean Productivity dataset. This dataset contains data from several oceanographic cruises accounting for roughly 3000
        production profiles. Accordingly, this dataset has been widely used to develop several models (Behrenfeld and Falkowski,
1997; Scardi, 2001) since it contains depth-resolved $^{14}$C phytoplankton production estimates coupled with chlorophyll *a*
        profiles, SST and PAR measurements. Such data are crucial for both studying phytoplankton production and developing model
        for estimating this process. Despite being a precious source of information, these field data cover only some ocean basins, they
        are affected by missing values and they have not been updated since 1994 (orange dots in Fig. 1). As the amount and quality
        of field data are paramount characteristics to understand the dynamics of natural processes, we wanted to create a new global
dataset expanding both the temporal and the spatial coverage of the previously cited one. Moreover, we decided to associate
        more production related information to each record, e.g. production to biomass ratio, bottom depth of the sampling station,
        distance from the coastline etc. The extra information could be extremely valuable for analysis and modelling purposes
        especially when machine learning techniques come into play (Peters et al., 2014; Recknagel, 2001).
        In order to retrieve phytoplankton production data, we consulted several sources which provide freely accessible information
such as PANGAEA, the Biological & Chemical Oceanography Data Management Office and the National Oceanic and
        Atmospheric Administration (a complete list of the exploited datasets with the respective references can be found in the
        supplementary materials).





To select suitable data, we adopted only four compulsory criteria that the newly found information had to meet. The first two criteria were related to the spatial-temporal context of the observations. Accordingly, we kept only the data for which date 65 (yyyy-mm-dd) and geographical coordinates (latitude and longitude) of the field measurements were recorded. The third fundamental requirement was the presence of depth-resolved $^{14}$C measurements, i.e. phytoplankton production profiles. Depth-resolved data are more informative with respect to the depth-integrated ones, since they provide not only information on the production magnitude but also on its vertical distribution. The final requirement was the measurement of chlorophyll *a* profiles associated to the production data. Even if several studies suggest that chlorophyll to carbon ratio could be extremely variable 70 depending on physical forces and phytoplankton physiological adaptation (Huot et al., 2007; Westberry et al., 2008), chlorophyll *a* is one of the most commonly used proxy for phytoplankton biomass, which in turn is a key parameter for studying the phytoplankton production. This is especially true when the relationship between the pigment and the biomass is not explicitly formulated, i.e. in the machine learning field. Furthermore, chlorophyll *a* can be easily measured with probes during sampling cruises and its surface concentration is also estimated from remote sensing platforms since 1978 (CZCS). The former 75 feature is important to exploit these measures to develop production models while the latter is crucial in a synoptic application of these estimators.

On the other hand, we did not discard records lacking other variables, such as SST or PAR. In fact, if these measurements were not available, we filled the gaps using interpolation techniques or retrieving the missing information from satellite platforms (see section 2.1).

Retrieving phytoplankton production data that were not present into the Ocean Productivity dataset and the gap filling operation allowed to expand both the spatial and the temporal coverage of this dataset. Spatial and temporal variability are important features in dealing with global assessments of natural processes such as phytoplankton primary production. The new dataset comprised 6084 production profile collected between 1958 and 2017, 2214 of which derived from the Ocean Productivity dataset. The need of larger amount of data related to phytoplankton production process was already highlighted by several 85 studies which either developed or compared primary production models (Campbell et al., 2002; Carr et al., 2006; Friedrichs et al., 2009; Lee et al., 2015; Saba et al., 2010; Scardi, 1996). In fact, from the latter type of studies emerged a high level of uncertainty in determining the global phytoplankton production. The range of estimated global production resulted from comparison papers was extremely large, highlighting how challenging could be modelling this process on a large scale.

Additionally, we enriched the new dataset with several qualitative and quantitative variables. These variables were either 90 derived from the existing one, retrieved from satellite platforms or extracted from freely accessible dataset (see section 2.2). Once the new dataset was structured, we highlighted its characteristics using several descriptive techniques and analysed the results from an ecological perspective (section 3).



## 2 Materials and methods

### 2.1 Data merging and reconstruction

As stated in the previous section, the most complete dataset of phytoplankton primary production was freely downloadable from the Ocean Productivity website. It contained roughly 3000 production profiles (Fig. 1, orange dots) associated with ancillary information such as chlorophyll profiles, SST and PAR measurements. We used this dataset as starting point and searched for data that could improve its spatial and temporal coverage. We conducted our searches mainly on PANGAEA and NOAA websites, which are freely accessible data repositories. Each dataset that we used in this work has been cited as specified

by the repository or the owners (see supplementary materials). We limited our search to datasets which contained depth-resolved measurements of net phytoplankton production as $^{14}$C associated with the respective chlorophyll $a$ concentration. The main reasons for this choice were the additional vertical distribution information provided by phytoplankton profiles with respect to depth-integrated estimates and the biomass proxy provided by the chlorophyll concentration. This feature allowed to analyse several characteristics of phytoplankton production, thus contributing to a deeper understanding of the whole

process.

The retrieved data were incorporated into the new dataset only if the geographic coordinates and the sampling date had been recorded. These data allowed to account for both the spatial and temporal variability of the phytoplankton production into the analysis.

For each retrieved dataset that met our requirements, the first step was to merge it with the Ocean Productivity one. From the

latter dataset we kept the following variables: Date of the sampling (yyyy-mm-dd), geographical coordinates of the sampling station (latitude and longitude, degrees), day length as hours of photoperiod (h), sampling depth (m), Pbopt (mg C mg Chla$^{-1}$ h$^{-1}$), SST (° C), surface PAR (E m$^{-2}$ day$^{-1}$), sampling depth chlorophyll $a$ concentration (mg m$^{-3}$), sampling depth daily primary production (mg C m$^{-3}$ day$^{-1}$) and integrated daily primary production (mg C m$^{-2}$ day$^{-1}$). To perform the merging procedure, we filled all the gaps in the newly retrieved data relative to the abovementioned variables. We computed the day length from the

latitude and the day of the year of the sampling. SST missing values were filled using MODIS daily data for observations from 2003 till present (MODIS-Aqua/Mapped/Daily/4km (nasa.gov)), multiple sensors daily data for records from 1981 to 2003 and the 1981-1990 mean for the data prior to 1981 (Copernicus SST) (Merchant et al., 2019). We used the MODIS values also for filling the PAR gaps from 2003 till present. The profiles previous to this date that lacked of PAR measures were discarded, since daily PAR estimates are available only through MODIS platform (late 2002-present). Discarding these data, the Ocean

Productivity dataset dropped from roughly 3000 profiles to 2214. We estimated the Pbopt parameter using the procedure proposed by Behrenfeld and Falkowski (1997a). Finally, we estimated the missing values in chlorophyll $a$ and primary production profiles with a depth-weighted average of adjacent values.





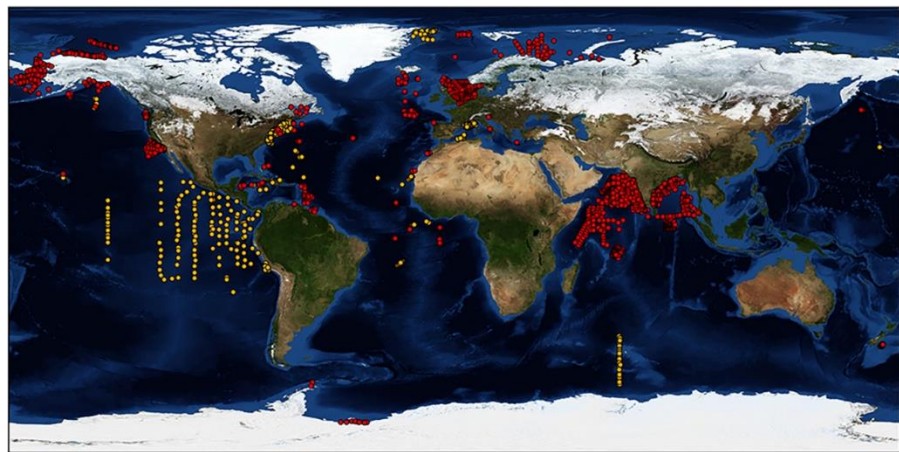

**Figure 1: Maps of the 6084 phytoplankton production profiles comprised in the new dataset. In orange the profiles derived from the Ocean Productivity dataset (2214) and in red the newly retrieved ones (3870).**

Once the merging procedure was finished, the new dataset contained 37722 records from 6084 profiles with respect to the 14300 and 2214 of the old one (Fig. 1).

## 2.2 Ancillary data association

We added to the dataset several variables related to phytoplankton primary production (table 1 and table 2). These variables can be divided in three groups: (i) data extracted from freely available datasets, (ii) numerical measures computed from the existing ones and (iii) categorical data derived from the previous two groups.

Among the variables that belong to the first group we list the bottom depth (m) and the stats related to it. We retrieved the bathymetry information from the GEBCO website (https://www.gebco.net/data_and_products/gridded_bathymetry_data/#global). We queried the GEBCO dataset using the geographic coordinates of the sampling stations to extract the bottom depth data. We also exploited up to 8 neighbour pixels to compute the bottom depth variance of the sampling point neighbourhood.

We retrieved information about the Mixed Layer Depth (MLD) using the Levitus model datasets (Levitus et al., 1994; Levitus and Boyer, 1994) which are freely available on the Levitus webpage (https://psl.noaa.gov/data/gridded/data.nodc.woa94.html). The last data that we gathered from an external dataset was the distance from the coastline (km). The 0.04 degrees distance dataset was downloaded from the NASA website (https://oceancolor.gsfc.nasa.gov/docs/distfromcoast/).

The second group of new variables was computed from information already present in the new production dataset at this stage. We computed the day of the year from the date, i.e. January the first and the last day of December were represented by 1 and 365 respectively.



| N | Variable Name | Short Name | Units | Method/Sensor |
|---|---|---|---|---|
| 1 | Count | / | / | / |
| 2 | Event | / | / | / |
| 3 | Short reference | / | / | / |
| 4 | Paper doi | / | / | / |
| 5 | Data doi/data link | / | / | / |
| 6 | Profile number | / | / | / |
| 7 | Date | / | yy/mm/dd | / |
| 8 | Year | / | / | / |
| 9 | Month | / | / | / |
| 10 | Day of the year | / | / | / |
| 11 | Latitude | / | degrees | / |
| 12 | Longitude | / | degrees | / |
| 13 | Day length | / | h | From day of the year and latitude |
| 14 | Bottom depth | / | m | GEBCO |
| 15 | Bottom depth standard deviation | Bottom depth sd | m | GEBCO |
| 16 | Mixed Layer Depth | MLD | m | Levitus et al. (1994); Levitus and Boyer (1994) |
| 17 | Distance from coastline | / | km | NASA website |
| 18 | Euphotic zone depth | Zeu | m | Morel and Berthon (1989) |
| 19 | Sampling depth | / | m | / |
| 20 | Max sampling depth | / | m | / |
| 21 | Max production depth | / | m | / |
| 22 | Sea Surface Temperature | SST | °C | *In situ* / MODIS-aqua |
| 23 | Surface Photosynthetic Active Radiation | Surface PAR | $E\ m^{-2}\ day^{-1}$ | *In situ* / MODIS-aqua |
| 24 | $Pb_{opt}$ | / | $mg\ C\ mg\ Chl a^{-1}\ h^{-1}$ | Behrenfeld and Falkowski (1997a) |
| 25 | Depth-resolved Chl *a* | / | $mg\ m^{-3}$ | *In situ* |
| 26 | Depth-integrated Chl *a* | / | $mg\ Chl\ a\ m^{-2}$ | Trapezoidal integration |
| 27 | Total Chl *a* | / | $mg\ Chl\ a\ m^{-2}$ | Morel and Berthon (1989) |
| 28 | Depth-resolved primary production | / | $mg\ C\ m^{-3}\ day^{-1}$ | *In situ* |
| 29 | Depth-integrated primary production | / | $mg\ C\ m^{-2}\ day^{-1}$ | Trapezoidal integration |
| 30 | Production to biomass ratio | P/B | $mg\ C\ day^{-1}\ /\ mg\ Chl\ a$ | / |

**Table 1: Production dataset numerical variables**



We also estimated the euphotic zone depth and the total chlorophyll *a* in the euphotic zone (mg chlorophyll *a* m$^{-2}$) using a
model developed by Morel and Berthon (1989).

Moreover, we extracted both the max sampling depth of non-null production value (m) and the depth at which maximum
production occurred for each profile (m) thus creating two new variables.

We estimated depth-integrated chlorophyll *a* and depth-Integrated Primary Production (IPP) by trapezoidal integration of in
situ measurements (mg chlorophyll *a* m$^{-2}$ and mg C m$^{-2}$ day$^{-1}$ respectively). Subsequently, we estimated the production to
biomass ratio dividing the depth-integrated phytoplankton production by the depth-integrated chlorophyll *a* concentration (mg
C day$^{-1}$ / mg chlorophyll *a*).

The last group of variables were generated by dividing the production profiles in classes on the basis of the previously
computed variables. We created the hemisphere variable assigning each profile to northern hemisphere, southern hemisphere
or equator on the basis of the sampling latitude. We also created a season variable on the basis of the date and northern
hemisphere season. We divided the year in four groups of three months each starting from January and tagged them as winter,
spring, summer and autumn respectively.

For numerical data, we applied the Jenks optimization algorithm (Jenks, 1967) to define the boundaries of six classes from
very low to huge (very low, low, moderate, high, very high, huge). Then we used these boundaries to assign each pattern to
one of the six classes. It is important to note this class segmentation is relative to our data rather than an absolute classification
criterion.

Finally, we investigated the relationship between the variables which are more intimately related with phytoplankton primary
production, i.e. SST, PAR, chlorophyll *a*, max sampling depth, max production depth and production to biomass ratio. We
produced heatmaps to provide an insight into the categorical variables and performed a Principal Component Analysis (PCA)
for their numerical counterparts.


| N | Variable Name | Short Name | Units | Method/Sensor |
|---|---|---|---|---|
| 31 | Hemisphere | / | / | / |
| 32 | Northern hemisphere season | / | / | / |
| 33 | Bottom depth magnitude | / | Very low to huge | Jenks (1967) |
| 34 | Bottom depth sd magnitude | / | Very low to huge | Jenks (1967) |
| 35 | Mixed Layer Depth magnitude | MLD magnitude | Very low to huge | Jenks (1967) |
| 36 | Distance from coastline magnitude | / | Very low to huge | Jenks (1967) |
| 37 | Euphotic zone depth magnitude | / | Very low to huge | Jenks (1967) |
| 38 | Max sampling depth magnitude | / | Very low to huge | Jenks (1967) |
| 39 | Max production depth magnitude | / | Very low to huge | Jenks (1967) |





| 40 | Sea Surface Temperature magnitude | SST magnitude | Very low to huge | Jenks (1967) |
|---|---|---|---|---|
| 41 | Surface Photosynthetic Active Radiation magnitude | Surface PAR magnitude | Very low to huge | Jenks (1967) |
| 42 | Pb$_{opt}$ magnitude | / | Very low to huge | Jenks (1967) |
| 43 | Surface Chl *a* magnitude | / | Very low to huge | Jenks (1967) |
| 44 | Depth-integrated Chl *a* magnitude | / | Very low to huge | Jenks (1967) |
| 45 | Total Chl *a* magnitude | / | Very low to huge | Jenks (1967) |
| 46 | Depth-integrated primary production magnitude | / | Very low to huge | Jenks (1967) |
| 47 | Production to biomass ratio magnitude | / | Very low to huge | Jenks (1967) |

**Table 2: Production dataset categorical variables**

**3 Results and discussion**

With this work we aimed at building a global phytoplankton production dataset updating the Ocean Productivity one.

Moreover, we wanted to expand the available information by associating several variables related to the primary production. The data underlying this article are available in the article online supplementary material.

The comprehension of natural phenomena deeply relies on available data. These complex processes often involve non-linear and not well-known relationships among their components. Accordingly, we believe that one crucial way to enhance our understanding of natural systems is provided by gathering information and then analysing it.

In this framework, we extended both the spatial and the temporal coverage of the Ocean Productivity dataset. These two features are paramount to boost our knowledge about the spatio-temporal distribution of phytoplankton production. In fact, the former allows to take into account temporal trends in the processes which are linked with climate related issues and food webs dynamics. The Ocean Productivity dataset contained data from cruises carried out between 1958 and 1994, which is a large span of time, but it has not been updated since then. Our data retrieval added 23422 new patterns from 3870 production profiles

which in most cases do not overlap with the Ocean Productivity temporal coverage. In fact, 2210 of the 3870 new phytoplankton profiles, i.e. roughly 57 % of the total, were collected between 1995 and 2017. Even if roughly 43 % of the new profiles shares the time coverage with the Ocean Productivity ones, the majority of these data does not overlap with the spatial coverage of the older dataset, thus enhancing the heterogeneity of the data.

Although the Ocean Productivity dataset was the most comprehensive source of information about phytoplankton primary

production, the bulk of its data was restricted to three main regions. These areas were the North-Western Atlantic, the Eastern Equatorial Pacific and the North-East Pacific along the West coast of the United States. The other ocean basins were under-sampled or not sampled at all (Fig. 1 orange markers). The new data improved the global coverage of the previous dataset.



Several profiles were added in the Arctic Ocean specifically in the Chukchi sea, the Beaufort sea, the Greenland sea, the North sea, the Norwegian sea, the Barents sea and the Kara sea. In the Pacific Ocean the new represented areas were the Bering sea,

the Gulf of Alaska, the areas off the Oregon and California coasts in addition to few production profiles gathered off the Eastern coast of New Zealand. In the Western Atlantic, new information was available for the Gulf of Saint Lawrence, the Florida coast and the Caribbean sea. In the Central Atlantic the new represented areas were located South-East off Ireland, South of Cape Verde island, off the Gulf of Guinea plus few records in the Bay of Biscay, off the coast of Morocco and in the Mediterranean sea. Few of the data in the Indian Ocean were present in the old dataset but we reconstructed missing

information and added new profiles from different datasets. The Southern Ocean remain strongly under-sampled, with the addition of few production profiles.

The temporal and spatial coverage of a dataset are crucial features. The first one allows to take into account the evolution of the studied process. This aspect is important in any type of assessment work, especially in a climate change context. In the phytoplankton production framework, the temporal span covered by the available in situ data could be used to study several

aspects. For example, repeated observations through the years for the same area could highlight temporal patterns of the investigated region. Moreover, this feature could be used to investigate the relationships between phytoplankton production and large-scale phenomena, e.g. El Nino-Southern Oscillation (ENSO). From a spatial perspective, the larger the global oceans area represented in the dataset the larger the spatial variability of the phytoplankton production process taken into account. This feature is crucial since both depth-resolved and depth-integrated phytoplankton production estimates are deeply

influenced by the geographic characteristics of the investigated area, e.g. latitude, distance from the coastline, bottom depth. Therefore, to deepen the understanding of this biological process we need to gather and analyse information from different areas. Finally, if we want to exploit a dataset to perform any global assessment on the phytoplankton primary production or tackle production-climate related issues we need an information pool that take into account as much variability of the process as possible (Behrenfeld et al., 2016; Gibert et al., 2018; Hays et al., 2005),.

One of the fields which heavily rely upon the amount and quality of the data is modelling. Several studies stressed how most of the limits in modelling the phytoplankton production depend upon the data availability (Campbell et al., 2002; Carr et al., 2006; Mattei and Scardi, 2020; Scardi, 2001). For these reasons, we believe that the enhancement of both spatial and temporal coverage of a freely available production dataset is an important contribution to modern oceanography.

We did not limit our work to homogenize several data source into a single one, but we also enhanced the amount of

phytoplankton-related available information. This type of information could be useful for boosting our understanding of primary production. Moreover, the ancillary data could be extremely valuable to model development, especially when machine learning techniques come into play. In fact, these approaches allow the use of variables as predictors even if the relationship with the target variable (primary production here) is not known (Catucci and Scardi, 2020; Franceschini et al., 2019; Olden et al., 2008; Peters et al., 2014; Recknagel, 2001).



The first two descriptors added to the new dataset were the hemisphere of the sampling station and sampling season, intended
       as northern hemisphere season. These two variables provided an insight into the global temporal and spatial distribution of the
       data (Fig. 2).

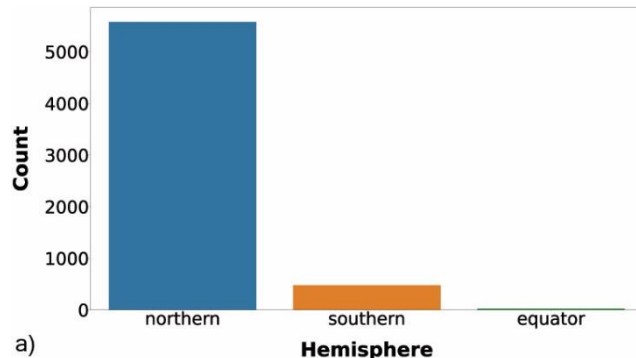 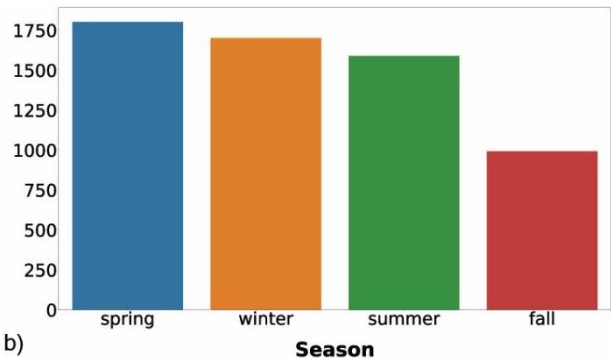

**Figure 2: a) Number of profiles gathered in the two hemispheres or at the Equator (5578, 478 and 28 respectively). b)**
**Number of profiles sampled in northern winter (January to April), Spring (March to June), Summer (July to**
       **September) and Fall (October to December).**

       The spatial distribution of the records was strongly unbalanced towards the northern hemisphere compared to the southern one
       (5578 vs 478 production profiles, Fig. 2a). This feature highlights the importance of gathering more data in the southern
hemisphere. In particular, the Southern Ocean is one of the least well-known areas of the global ocean and the uncertainty
       related to this lack of knowledge negatively affects our understanding of both the global phytoplankton production and carbon
       cycle (Arrigo et al., 2008; Caldeira and Duffy, 2000; Moigne et al., 2016; Reuer et al., 2007).
       On the other hand, the temporal variability in the new dataset is more balanced with respect to the spatial one. Accordingly,
       the number of profiles sampled during the northern Winter, Spring, Summer and Fall are respectively 1701, 1802, 1589 and
992. This is an important feature especially for the areas characterized by seasonal patterns which not only influence the
       magnitude of primary production but also its distribution along the water column (Falkowski and Raven, 2007a). Therefore,
       when both the depth-integrated and the depth-resolved perspectives are taken into account, this temporal variability is doubly
       valuable.
       We also added information related to the bathymetry of the sampling area. We queried the GEBCO dataset to extract the
bottom depth of sampling stations. Afterwards, we applied the Jenks optimization algorithm to partition the data into six classes
       (Fig. 3).



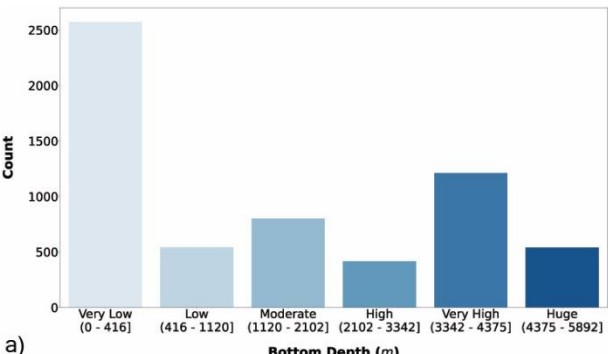 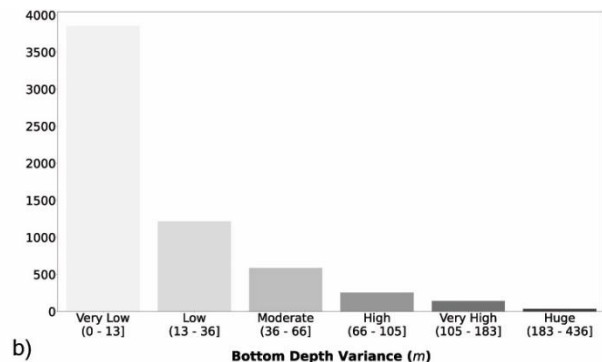

**Figure 3: a) Bottom depth and b) its variance classes. The bar colour intensity reflects the magnitude of the class values.**


The majority of the observations were collected in areas shallower than 416 m (Fig. 3a). This feature highlights that the continental shelf areas are the most frequently sampled ones. The second class in terms of abundance was the very high one, while the other classes had less than 1000 profiles each. In Fig. 3b we can notice that almost all the sampling stations had a very low bottom depth variance in their neighbourhood, thus the area of the sampling was homogenously deep. Bathymetry

related information could help understanding the geomorphological region of the ocean where the sampling station was situated, i.e. coastal, continental shelf or open ocean.

The depth information could help us analysing the profiles characteristics, since it could be interpreted as a proxy for several features such as nutrients availability and water column dynamics. In fact, even if the depth is not directly related to the phytoplankton production, it is an important physical descriptor of the ocean system in which this biological process occurs.

The MLD data were retrieved from the Levitus dataset. These estimates provide a seasonal indication for the water column mixing status which is related to both the magnitude and the vertical distribution of the phytoplankton production. We also added the distance from coastline as ancillary information. This distance provides an insight into how much factors like terrestrial runoff, rivers and waste water discharges could affect the primary production. It is well known that coastal areas are characterized by higher level of primary production mainly due to nutrients inputs from natural and anthropogenic sources

(Paerl et al., 1990; Teixeira et al., 2018; Wollast, 1998).





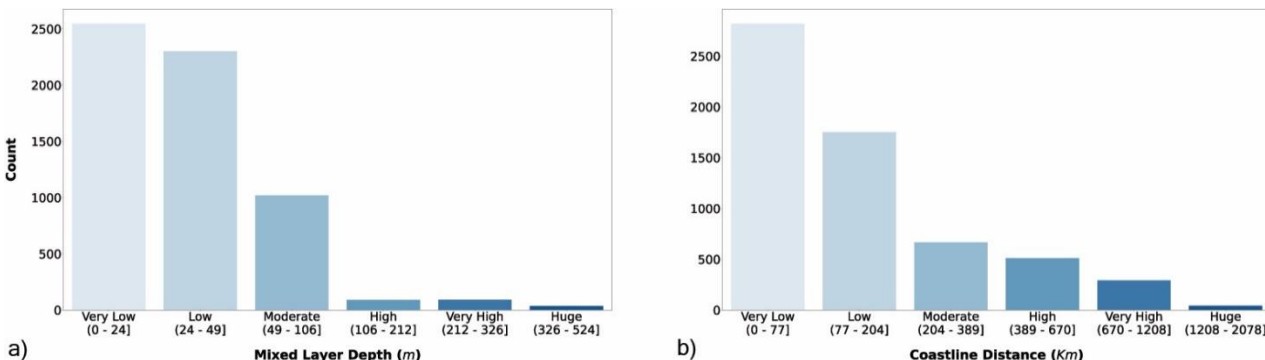

**Figure 4: a) MLD (m) and coastal distance (km) magnitude. The bar colour intensity reflects the magnitude of the class values.**

Figure 4a shows that 96.4 % of the sampling stations presented a very low to moderate MLD. The distance from the coastline showed the same pattern with the bulk of the profiles comprised in the first two classes (Fig. 4b). The main reason for adding these variables to our dataset is their relationship with nutrient availability which generally became scarcer as the distance from the coastline and the bottom depth augment. Moreover, the available nutrients are distributed in different concentrations along the water column according to the MLD magnitude (Falkowski and Raven, 2007a; Huisman and Weissing, 1995; Jäger

et al., 2008). The latter feature is one of the factors influencing the vertical distribution of the phytoplankton production. Another group of variables was extracted directly from the sampling data. We created the maximum sampling depth as the depth at which the deepest water sample was collected (Fig. 5a). Usually, this depth corresponds to the 1% of the surface irradiance, but it was not specified in all the retrieved data. We also introduced the maximum production depth which is the depth where the maximum depth-resolved production value occurred, i.e. the peak of the production profile (Fig. 5b).


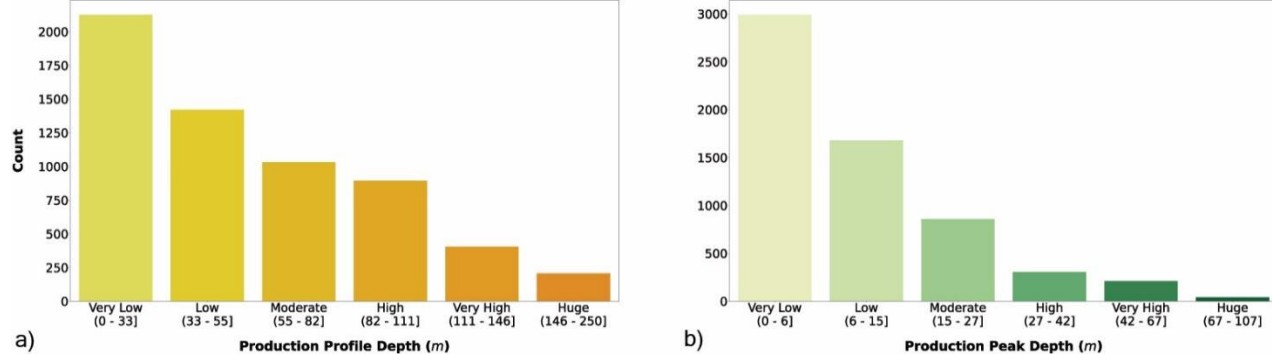

**Figure 5: a) Maximum profile depth and b) maximum production depth classes. The bar colour intensity reflects the magnitude of the class values.**

The majority of the records showed from very low to high production profile depth. In fact, these 4 classes included 94.3 % of the records. Among these classes the most represented was the very low one (till 33 m). This feature reflected again the higher



number of coastal profiles with respect to the open ocean ones. The profile peak depth showed an even stronger decreasing trend with respect to profile depth. In fact, 76.7 % of the patterns were characterized by a peak comprised in the first two classes. The decrease of primary production with depth is mainly justified by the light attenuation along the water column

which is one of the main physical forcing for phytoplankton production. In fact, even if deeper waters are usually nutrient-rich while the shallower ones are nutrient-depleted, the photosynthetic process cannot prescind from light availability.

SST and surface PAR variables were already present in the Ocean Productivity dataset but showed several missing data. As described in the section 2.1, we filled the gaps where possible in both the old and the new data. The results of the Jenks algorithm on SST and PAR variables are presented in Fig. 6.


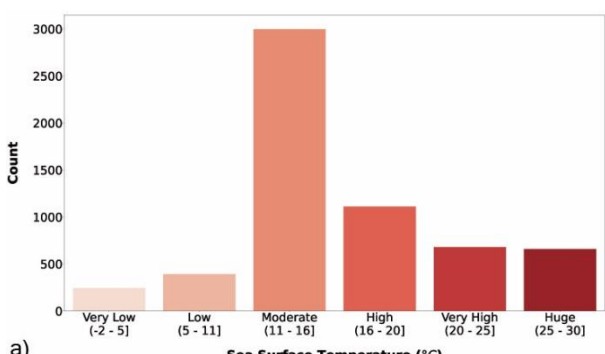
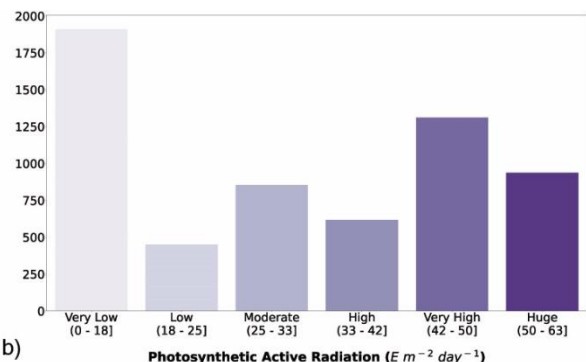

**Figure 6: a) Sea surface temperature and b) photosynthetic active radiation classes. The bar colour intensity reflects the magnitude of the class values.**

SST and surface PAR showed different patterns with respect to the previously discussed variables. The bulk of the records showed moderate values of SST (roughly 50 % of the production profiles), while the lowest and highest values were the less abundant. The surface PAR classification showed a different pattern in which the very low values were the majority followed by the very high and the huge values.

These two parameters exert an important influence on phytoplankton primary production and they have been key factors in

modelling this biological process. In fact, SST affects physiological characteristics of phytoplankton influencing its primary productivity and PAR represents the share of solar energy that is used for $CO_2$ fixation.

Unfortunately, most of the times these parameters are measured only at surface level while it could be extremely useful to have depth-resolved in situ measurements for studying phytoplankton production from a depth-resolved perspective.

One of the most important variables related to phytoplankton production is the depth-resolved chlorophyll *a* concentration. It

was one of the compulsory requirements for inclusion in the gathered data set. Even if the relationship is not straightforward, it is often used as phytoplankton biomass proxy. Several works pointed out that other variables could be a more precise proxy (Huot et al., 2007; Westberry et al., 2008), but it is often difficult if not impossible to compute them for old data, thus limiting the effectiveness of the new candidates.

Starting from the chlorophyll *a* profiles, we also computed the depth-integrated values using a trapezoidal integration. We

exploited the depth-integrated value to compute a production to biomass ratio and as source of information for the dataset analysis.

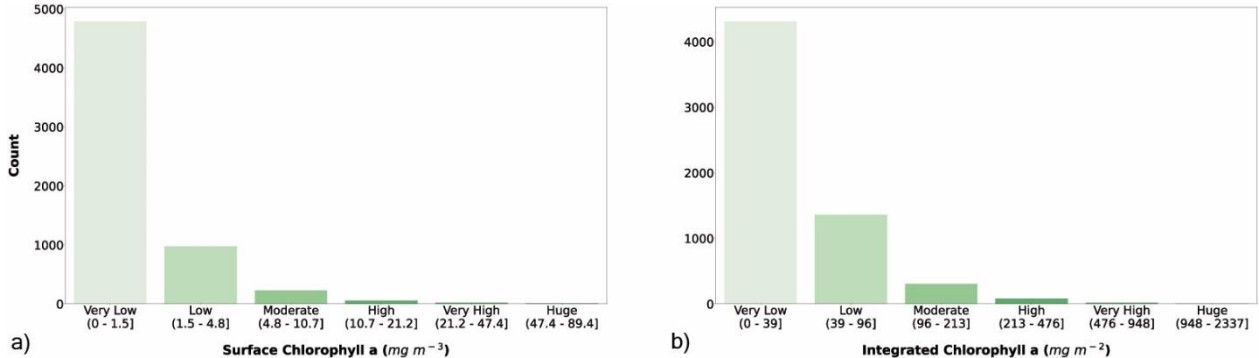

**Figure 7: a) Surface chlorophyll *a* and b) depth-integrated chlorophyll *a* classes. The bar colour intensity reflects the magnitude of**

**the class values.**

The classification of surface chlorophyll *a* concentration (Fig 7a) showed that 78.7 % of the phytoplankton profiles in the dataset fell in the very low class and the first three classes comprised 98.5 % of the records. Surface chlorophyll *a* concentration is one of the main variables used to predict phytoplankton production since it is related to the biomass of these autotrophic

organisms. Moreover, this variable is retrievable through remote sensing platforms thus allowing a quasi-synoptic application of production estimators.

The segmentation of the integrated chlorophyll *a* concentration (Fig. 7b) showed a similar pattern compared to the surface one. In fact, the first three classes were the most abundant (98.2 %), but Fig. 7b shows a larger number of low and moderate values than Fig. 7a (27.4 % vs 19.8 %).

We considered the availability of phytoplankton production profiles as compulsory information for the newly retrieved data. The reason for this requirement was twofold: firstly, we wanted to keep all the information already present in the Ocean Productivity dataset, which contained depth-resolved measurements of phytoplankton production. Secondly, we believed that the study of phytoplankton production could benefit from the coupled information of magnitude and its distribution along the water column with respect to taking into account only the former. Starting from the depth-resolved production data (mg C m$^{-3}$ day$^{-1}$), we computed the depth-integrated production using a trapezoidal integration (mg C m$^{-2}$ day$^{-1}$). Subsequently, we

computed a production to biomass ratio using depth-integrated phytoplankton production and depth-integrated chlorophyll *a*. The segmentation in classes of IPP and production to biomass ratio is shown in Fig. 8.



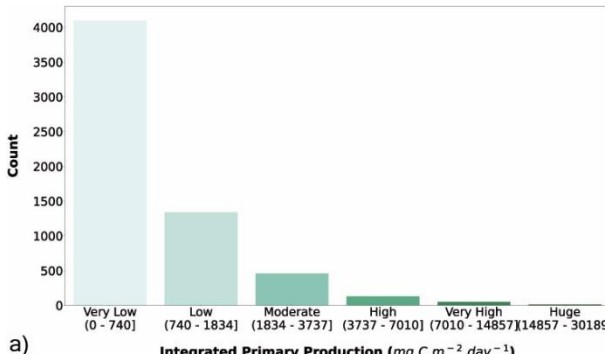 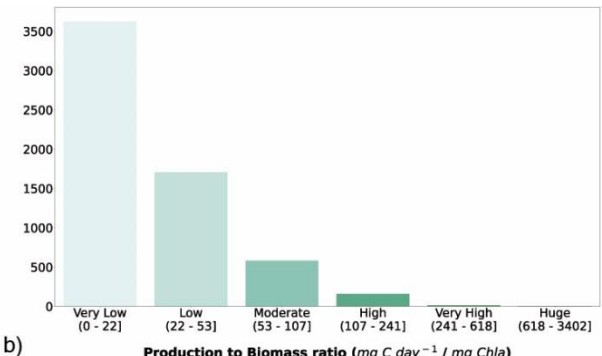

**Figure 8: a) Depth-integrated phytoplankton primary production b) production to biomass ratio classes. The bar colour intensity reflects the magnitude of the class values.**

Both IPP and production to biomass ratio classifications showed the same pattern. Accordingly, the larger the class values the lesser the numerosity of the class. The first class comprised 67.3 % and 59.5 % of the profiles for IPP and production to

biomass ratio respectively.

IPP is an important measure in global assessments of phytoplankton production. It provides a bidimensional view (latitude vs longitude) of the oceanic production which in turn influences several biological and non-biological processes in the biosphere, e.g. energy flow in the marine food webs, fish landings and $CO_2$ absorption (Anderson et al., 2018; Barange et al., 2014; Blanchard et al., 2012b; Caldeira and Duffy, 2000; Carvalho et al., 2017; Kwak and Park, 2020b; Maureaud et al., 2017; Shurin

Jonathan B et al., 2006). On the other hand, depth-resolved production provides more insights into the phytoplankton production process characteristics which in turn could lead to better estimates of IPP (Mattei et al., 2018).

Production to biomass ratio could convoy valuable information on the physiological state of the phytoplankton which in turn is influenced by biotic and abiotic forcing. This ratio can be also used to further analyse the profiles characteristic and to decide whether they are suitable or not for specific purposes, e.g. modelling phytoplankton primary production (Mattei and Scardi,

2020; Scardi, 2001).

Subsequently, we selected a subset of these variables and described their relationships with the depth-integrated phytoplankton production (see heatmaps Fig. 9 to 15).



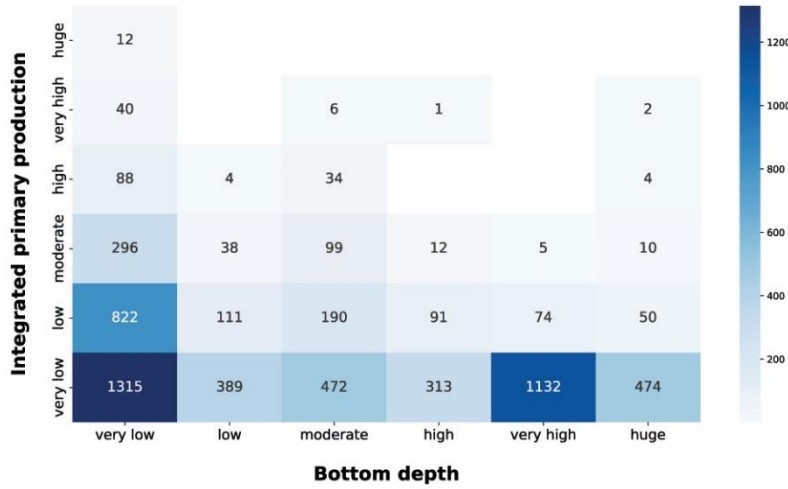

**Figure 9: IPP vs Bottom depth. The blue heatmap highlight the difference in production potential between coastal and open ocean areas.**

In the integrated production vs bottom depth, the very low production class was the most abundant in all the depth ranges (Fig. 9). This feature was prominent in the very low and very high bottom depth classes in which comprised roughly 60 % of the very low production profiles. In very shallow areas the production could be limited to a small portion of the water column thus often resulting in low integrated production values. On the other hand, open ocean areas are usually nutrient depleted thus phytoplankton production is limited even if other environmental conditions are favourable. Shallower sampled areas showed higher levels of depth-integrated production. This was manifest for the very low class, which showed a noticeable amount of profiles for each production class and the bulk of largest depth-integrated values i.e. 67.7 %, 81.6 %, 100 % of the high, very high and huge production profiles respectively. The latter feature was mainly due to land inputs to coastal areas which, when associated to favourable physical conditions, lead to high production levels. The blue heatmap highlighted the high potential of shallower areas in contrast with the low one of the open ocean zones.

The grey heatmap complement the information of the blue one taking into account the local variance of the bottom depth (Fig. 10).





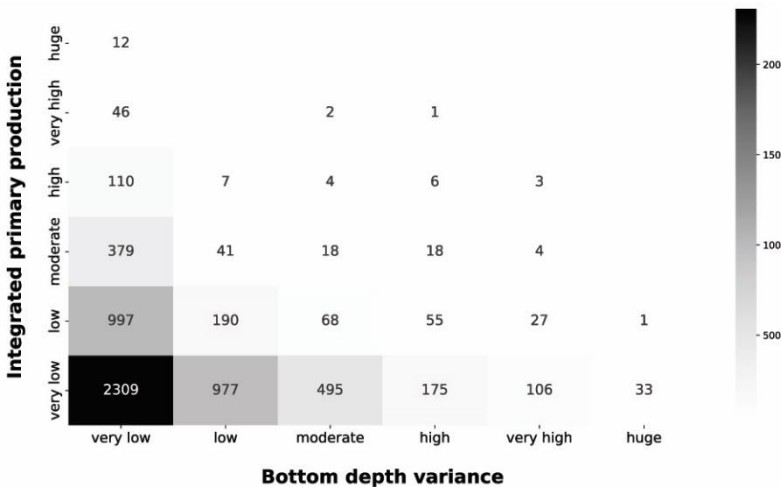

**Figure 10: IPP vs Bottom depth variance. The grey heatmap shows the relationship between the variance of the bottom depth and the IPP.**

The very low bottom depth variance comprises both very low and huge production profile. The low level of variance characterizes coastal areas, in which bottom depth is consistently low, and the open ocean zones, in which the bottom depth was consistently high. Progressively larger variance values showed the transition from shallower to deeper areas which corresponds to a decrease in depth-integrated production. This is consistent with our previous analysis and with phytoplankton ecology.

Subsequently, we analysed the relationship between integrated production and the profile depth (Fig. 11).

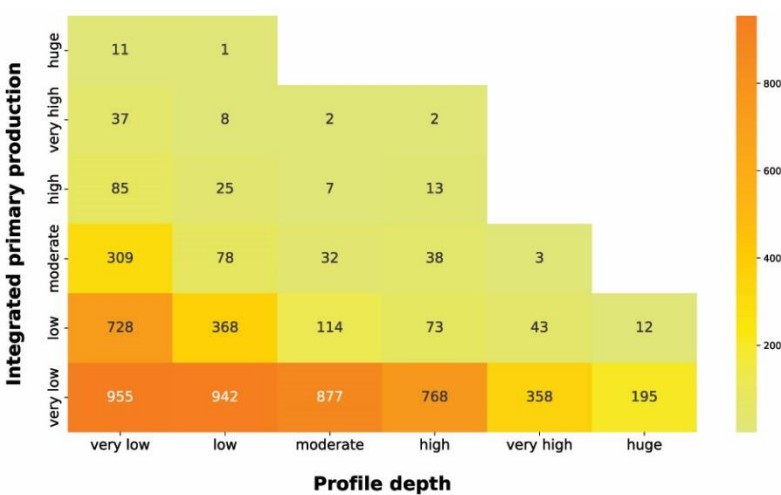

**Figure 11: IPP vs Profile depth. The yellow to orange heatmap highlights the relationship between the depth-integrated phytoplankton production and the production profile depth.**






The yellow to orange heatmap showed that the bulk of high production profiles was comprised in the first two profile depth classes. Shallow production profiles are usually the ones closer to the coastline or upwelling zones. These areas are nutrient-rich even in surface waters, where light availability is high, thus allowing high level of production. Moreover, high levels of production in shallow waters enhance the light attenuation phenomenon reducing the column water area suitable for primary

production. Conversely, the deeper the phytoplankton profile the lower the depth-integrated production. Low-nutrients conditions lead to low phytoplankton biomasses values and thus to a deeper light penetration along the water column. The latter feature allows the structuring of deeper production profiles. Although these profiles occupy a large portion of the water column, the total profile production is limited by the scarce nutrients level.

Continuing our analysis of the relationship between depth-resolved features and magnitude of depth-integrated production, we

took into account the production peak depth (Fig. 12).

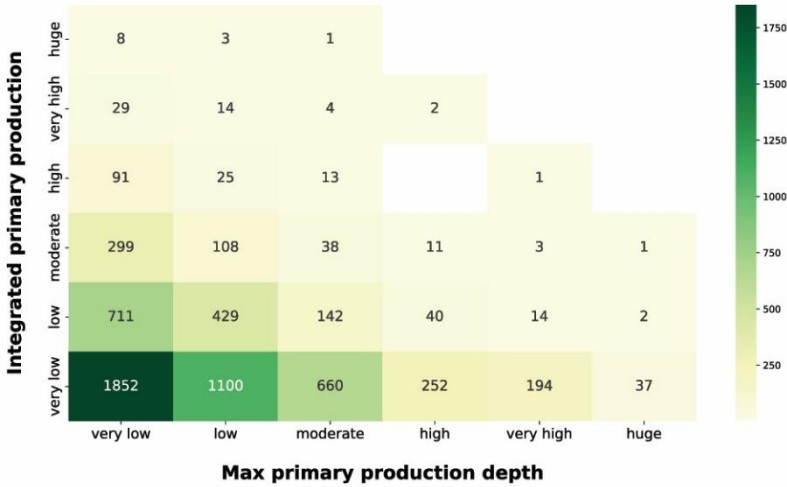

**Figure 12: IPP vs maximum primary production depth. The yellow to green heatmap shows how the magnitude of IPP is related to the depth at which the maximum production occurs.**


The production distribution along the water column is influenced by physiological and physical forcing. The optimum between light and nutrient availability determines the depth at which the maximum production occurs (Falkowski and Raven, 2007b). Since light availability exponentially decrease with depth, shallow peaks reflect either a condition of low irradiance or high irradiance and high nutrients. Both these situations lead to surface production peaks which are associated with a wide range of

integrated production magnitudes. The yellow to green heatmap highlighted how the high depth-integrated magnitudes are associated only with shallow peak profiles. This feature reflected the relationship between phytoplankton physiological needs and the light extinction behaviour. Deep production peaks indicate a nutrient paucity condition in shallow waters which shifts the optimum condition near the nutricline depth. From the integrated production perspective, low values were associated with



shallow peaks in conditions of low PAR or low nutrients even in deeper areas of the water column. Highest levels of production
were coupled with surface or sub-surface peaks, while deeper peaks (high to huge) represented 9 % of the total profiles and
showed only very low to moderate depth-integrated production with the exception of 3 production profiles.

Among the physical forcing that influences the phytoplankton production we explored the characteristics of SST and PAR
(Fig. 13 and 14). It is worth stressing that the segmentation derived from the Jenks algorithm is relative to our data. For
instance, the procedure was influenced by the under representation of circumpolar areas especially in the colder months.


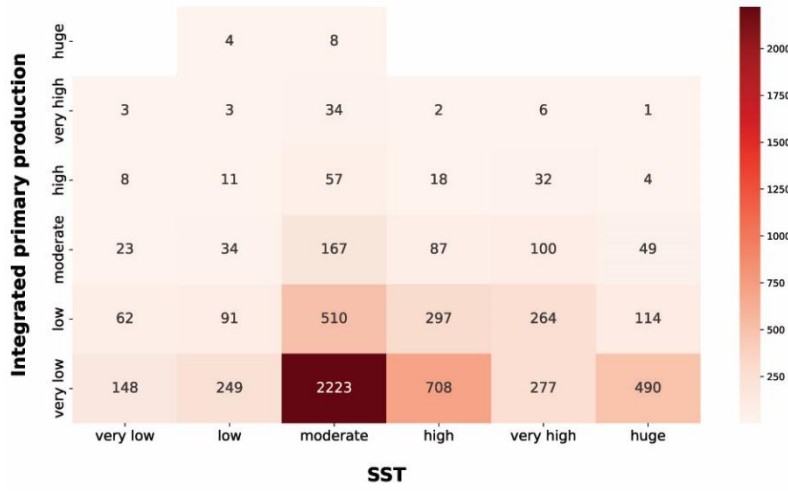

Figure 13: IPP vs SST. The red heatmap represents the relationship between depth-integrated production and SST.

The red heatmap showed that very low and low levels of SST were associated mainly with low primary production magnitude.
The same pattern characterized very high and huge levels of SST. These features are related to primary production seasonality
induced by physical forcing. The former situation referred to cold seasons in which the nutrient levels in the water column is
high but not enough solar radiation is available for the photosynthetic organisms. The latter reflects a strong shallow
stratification of the water column which is typical of warm seasons or areas constantly subjected to high levels of irradiance.
This leads to low nutrient concentration in shallow waters which in turn severely limits the primary producers. Moderate levels
of SST were associated with a wider range of values and comprised the larger levels of phytoplankton production. This feature
could be associated with the transition between cold and warm seasons. In this period of the year the environmental conditions
are optimal for primary production since the high nutrient concentration accumulated during the cold season became
exploitable due to the increasing available solar radiation.


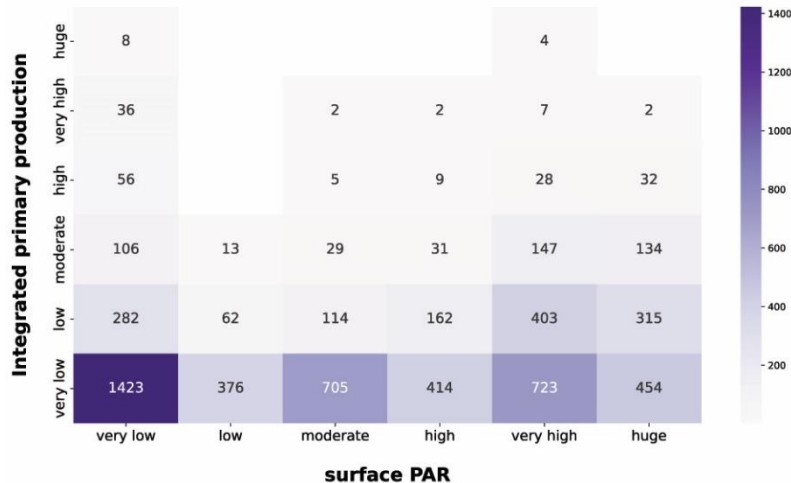


**Figure 14: IPP vs PAR. The purple heatmap shows how integrated phytoplankton production and PAR are related to each other.**

The first feature highlighted by the purple heatmap (Fig. 14) was the large share of very low integrated production profiles in each PAR class. This is mainly related to the nutrient availability, since low nutrients concentration leads to very low

production levels independently from the physical forcing.

Not surprisingly, very high and huge levels of PAR were associated with larger magnitudes of integrated production since the photosynthesis is intimately related to the solar radiation.

Another striking aspect was the wide range of phytoplankton responses to very low PAR magnitudes. In fact, all the production levels are well represented in this PAR class showing that the geographical characteristics of the area deeply influences the

primary producers. Accordingly, a constant nutrient input from terrestrial run-off can boost the primary production especially in shallower layers of the water column where usually it is nutrient limited.

The last relationship that we analysed was the one between IPP and depth-integrated chlorophyll *a* (Fig 15).





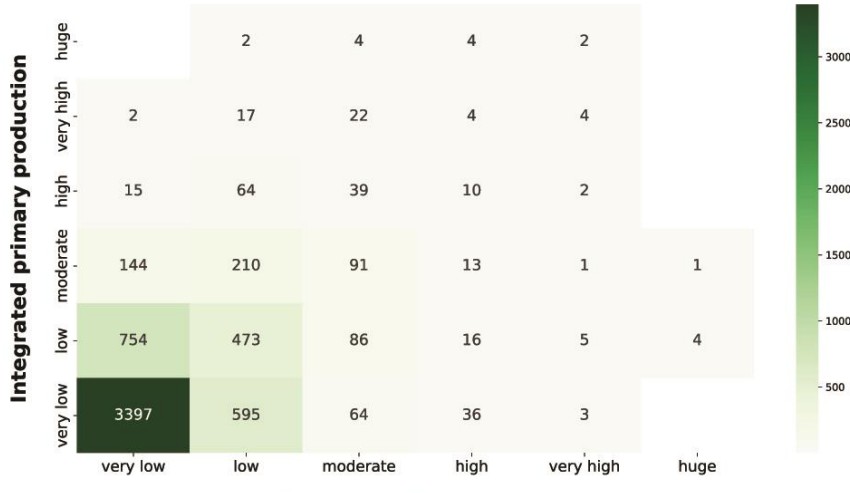

**Figure 15: IPP vs chlorophyll *a*. The green heatmap relate the depth-integrated phytoplankton production to the chlorophyll *a* magnitude which is one of the most used proxy for phytoplankton biomass.**

The pattern emerged from the green heatmap (Fig. 15) was a proportional one till moderate chlorophyll *a* values. Accordingly, the higher the integrated production the higher the integrated chlorophyll *a*. The bulk of the profiles was comprised in the very low and low integrated chlorophyll *a* classes. 3992 profiles (65.6 % of the total patterns) from very low and low chlorophyll *a* concentration were coupled with very low production, while 1227 (20.1 % of the total patterns) were associated to low production. Conversely, higher levels of integrated chlorophyll *a* were characterized by a larger share of high production profiles. This was not surprising since the chlorophyll *a* is the principal photosynthetic pigment and its raise is caused by physiological needs of phytoplankton or biomass augmentation.

The final analysis we carried out was a PCA to spot and analyse general patterns in the dataset. We selected the following twelve variables to perform the PCA: day length, bottom depth, bottom depth variance, MLD, distance from coastline, SST, PAR, surface chlorophyll *a*, integrated chlorophyll *a*, surface phytoplankton production, integrated phytoplankton production and production to biomass ratio (Fig 16).

We used the type one scaling since our main focus was on the position of the profiles. Using this type of scaling the distance between the objects in the plot approximate their Euclidean distances in full dimensional space. The variance explained by the first and the second axis was 0.26 and 0.15 respectively. The relatively low share of explained variance highlights the high complexity of the data which encompass large levels of spatial and temporal variability. Nevertheless, the ordination allowed to spot and show several features of the production dataset. From a general point of view (Fig. 16a) we can see that high levels of surface and depth-integrated chlorophyll *a* are associated between them. The same remark is valid for the phytoplankton production. Moreover, is not surprising that chlorophyll *a* concentration and phytoplankton production measures point in the



same direction along the first axis. Another feature that is consistent with the results previously presented was the inverse relationship between bottom depth and coastline distance with respect to primary production magnitude.

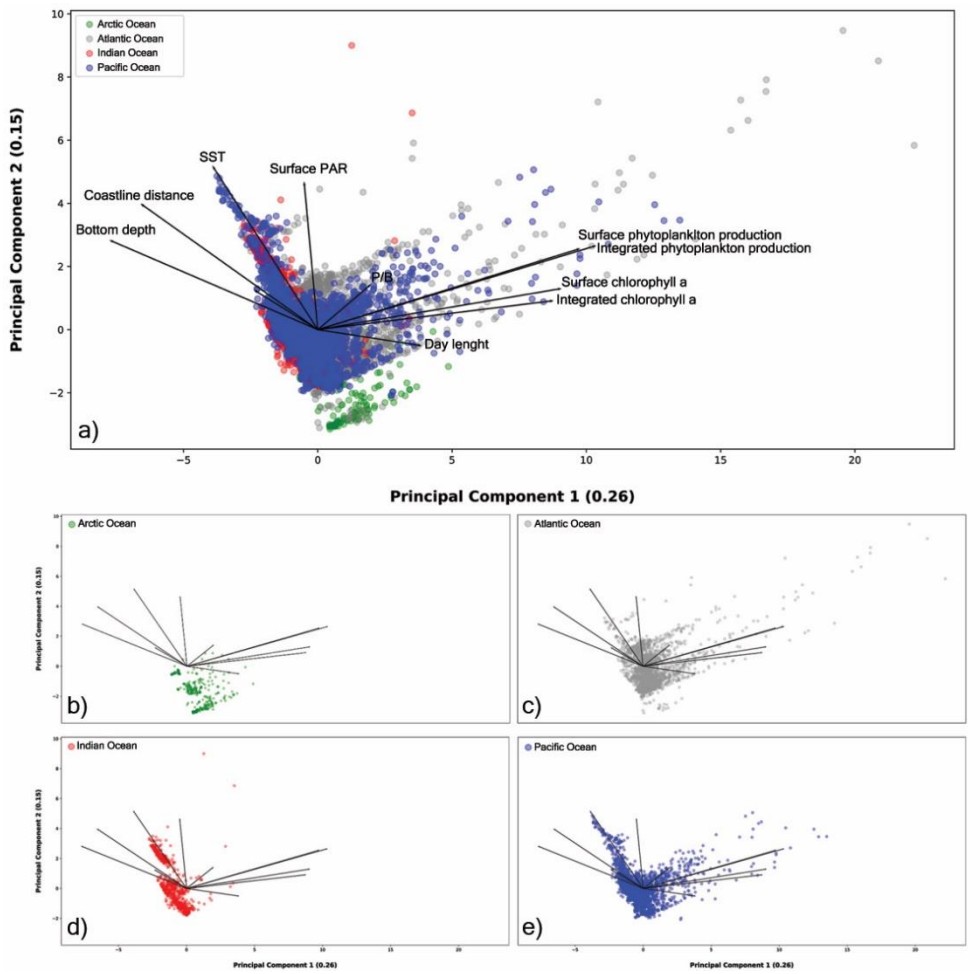

**Figure 16: Principal component analysis 0.26 & 0.15 explained variance from the first two axes respectively. The explained variance from the first two components was equal to 0.41. a) global PCA; b) Arctic Ocean profiles; c) Atlantic Ocean profiles; d) Indian Ocean profiles; e) Pacific Ocean profiles.**

The Arctic Ocean (Fig. 16b) presented the narrowest cloud of point among the basins. This could be the result of the low
number of records collected in this region and the peculiar characteristic of the area. In fact, the Arctic Ocean is characterized by low level of SST and PAR throughout the year with the exception of short periods of time.

The Indian Ocean showed two groups of samples. This feature was the result of the monsoon system that characterize this basin. The wind blows from northeast during cooler months and from southwest during the warmest months of the year (Dickson et al., 2001). Moreover, the plot (Fig. 16d) shows that this is not a highly productive area independently from the

485 environmental conditions. In fact, the bulk of the points were placed in the opposite direction of both chlorophyll *a* concentration and phytoplankton production levels.

A large amount of information was associated with the Atlantic and Pacific Oceans (Fig. 16c and 16e respectively) since they were the most sampled areas. Accordingly, they showed the largest range of sampled conditions with large and low levels for almost every environmental and biological variable. Moreover, almost every profile associated with a high level of

490 phytoplankton production or chlorophyll *a* concentration was recorded in these basins.

## 4 Data availability

The dataset described in this work is published in the PANGAEA repository (DOI: https://doi.pangaea.de/10.1594/PANGAEA.932417) (Mattei and Scardi, 2021) and it will remain protected until this work will be published. A pdf file containing supplementary data information is available in the data repository. The anonymous access

495 link for the review process is the following: https://www.pangaea.de/tok/af75beb4e8e2be6577041b4ec49eb91fb9b82c82.

## 5 Conclusions

The data paucity is one of the most important issues related to several disciplines and Ecology makes no exception. This is especially true if the task to tackle is the understanding of the dynamics of a complex biological process, such as phytoplankton primary production, on a global scale. Moreover, several researchers during the last decades highlighted how the lack of data

500 is the main constraint for modelling phytoplankton production.

In this framework, we believe that building a new, homogenous and ready to use dataset, associated with a general analysis of its features, could play an important role in the study of phytoplankton production. For this reason, we retrieved phytoplankton production data from heterogeneous sources and created a new global dataset. We also applied several data analysis and visualization technique to spot and discuss both the dataset characteristics and the variables relationships.

505 Furthermore, enriching the dataset with ancillary data related to the phytoplankton production could be extremely useful in improving our understanding of this pivotal process, e.g. in a machine learning context.

Despite the new dataset is still unbalanced from a spatial and temporal perspective and the need for new data will never be fully satisfied, we believe that it represents a crucial improvement of the previous existing ones.

## Author contribution

510 FM collected, processed and analysed the data. FM wrote the paper. MS supervised the whole work.



**Competing interests**

The authors declare that they have no conflict of interest.

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
