# Peer review of "Collection and analysis of a global marine phytoplankton primary production dataset"

_Earth System Science Data, 2021_

## Author Response (AR1)

We would like to thank both the reviewers for the time spent to provide these constructive observations and for recognising the effort behind the proposed work. We addressed all the points raised by the reviewers and we modified the manuscript and the data accordingly.

Reply for the Anonymous Referee #1

**1) My suggestion to the authors is to either flag the data that have been filled in with a separate product, or to not fill-in this data at all. I would lean more towards the latter, as many data users will already have a specific data product in mind for a variable like SST, that may work better for their region or application of interest. This also allows the user to have a clear understanding of the uncertainty surrounding that SST estimate too.**
**PAR is another example of the previously mentioned issue, which contains a blend of in-situ and satellite-derived values.**
We thank both the reviewers for this remark. We opted to add a flag column for both SST and PAR to clarify whether the value was an *in situ* (flag = 0) or a reconstructed one (flag = 1).

**2) Additionally, the authors actually discarded nearly 800 profiles due to the lack of an in-situ or remotely sensed PAR estimate. I don't follow the reasoning for why these data were discarded considering that PAR was not one of the four criteria that the authors established for generating the dataset.**
We thank the referee for this observation. We discarded the abovementioned profiles since we wanted to provide a ready to use dataset with no missing values. The excluded data were among the oldest ones and for this reason the PAR imputation procedure failed. Moreover, the spatial coverage of these data overlaps with more recent records included in the dataset. For these reasons, we opted for removing the data.

**3) Apart from these concerns with SST and PAR, there are some other variables that seem more extraneous and in my opinion would be better left for the user to define, if necessary. For example, the Northern hemisphere seasonal classification is not applicable for the Arctic, where a significant portion of the data are located. This variable is probably best left for the user to define based on their specific application. The Jenks (1967) data classification schemes are useful for generating the manuscript figures that illustrate data distribution, though I'm not sure how useful these variables will be for other users. As the authors describe in the text, the classifications are generated specifically for the entire dataset, meaning that they will change if a user selects a subset of the data.**
We thank the reviewer for this observation. We believe that the ancillary variables attached to the dataset could be extremely valuable, especially when machine learning techniques for modelling, pattern recognition and data mining come into play. The approaches of this computer science branch can exploit the relationships between variables without any *a priori* knowledge, thus enhancing the value of any information directly or indirectly related to the target of a specific work, i.e. phytoplankton primary production. We believe that leaving to the user the decision to exclude any variable that does not fit his/her purpose is the optimal solution in this case.

**4) In the introduction, I suggest that the authors provide a little more background on the distinction between chlorophyll a and primary productivity. The authors allude to important differences between the two variables but do not go into much detail. This is especially relevant in order to drive the motivation behind this dataset, since productivity data are generally more scarce than Chlorophyll.**
We thank the referee for this suggestion. We added the following paragraph in the introduction:" Chlorophyll *a* is the most abundant pigment in photosynthetic organisms and it is responsible for the light energy absorption. The concentration of this pigment is intimately related to phytoplankton

productivity, i.e. the production of organic matter. In fact, the energy gathered from sunlight allow to fix carbon dioxide into matter" (from line 69 to 71).

**5) Line 26: Change to "… of global productivity"**
We applied the correction suggested by the reviewer.

**6) Line 60: Here and in the supplement, please be more specific than just "The National Oceanic and Atmospheric Administration" regarding where you pulled the data from. NOAA is a large entity and it's not clear where these data are virtually located.**
We thank the referee for the observation. We specified that the data were downloaded from the National Centers for Environmental Information.

**7) Line 74: What is "CZCS"?**
We thank the reviewer for this remark. We clarified that the CZCS acronym refers to the Coastal Zone Color Scanner (https://oceancolor.gsfc.nasa.gov/data/czcs/).

**8) I would think the "Conclusions" section should come before the "Data Availability" section at the end of the manuscript, unless this specific format required by the journal.**
This is the specific format required by the journal.

**9) Figure 2 caption: Should the seasonal definitions be winter (January to March), spring (April to June)…?**
We thank the reviewer for spotting this error in the figure caption. We have corrected the mistake.

Reply for the Anonymous Referee #2

**10) I agree with the comments of referee 1 regarding the lack of a clear information of the origin of some data; it is crucial for the reader to understand if the data is from in situ or obtained by modelling.**
See point 1 Referee #1.

**11) However, the text based on Figure 16 seems a bit poor. Also, graphs here are too small. The distinction of the different oceans could perhaps be more developed.**
We thank the reviewer for this observation. We split Figure 16 in two different figures. Figure 16 now shows only the complete PCA, while figure 17 is an enlarged representation of the four Oceans specific points. We also enriched the figures captions that were indeed a bit poor and added few comments on the different oceans characteristics.

**12) The paper could establish links with previous papers which gathered PP parameters such as Bouman et al, Earth Syst. Sci. Data, 10, 251–266, 2018 https://doi.org/10.5194/essd-10-251-2018 and Kulk et al Remote Sens. 2020, 12, 826; doi:10.3390/rs12050826, which as a similar scope, but with satellite data.**
We thank the reviewer for highlighting how the proposed work can enrich the available information on the global phytoplankton primary production especially if combined with related published products. We made evident this connection and its importance for understanding the primary production process in the conclusion section.

---

## Author Response (AR2)

**My only other suggestion is to move the location of these flags in the dataset to immediate follow the variable that they describe. Currently, they are placed at the very end which is far removed from the SST and PAR variables the flags accompany.**

We would like to thank the editor for his work and its suggestion. We moved the flag columns right next to the variables they refer to. We updated the tables in the manuscript to reflect the changes in the dataset.